# Knowledge, attitude, and practice of pharmacy professionals on generic medicines in Eastern Ethiopia: A cross-sectional study

**Ammas Siraj Mohammed** [1]*, **Nigist Alemayehu Woldekidan**[2], **Fuad Adem Mohammed**[1]

**1** Department of Clinical Pharmacy, School of Pharmacy, College of Health and Medical Sciences, Haramaya University, Haramaya, Ethiopia, **2** School of Pharmacy, College of Medicine and Health Sciences, University of Gondar, Gondar, Ethiopia

* Ammassiraj2337@gmail.com

## Abstract

### Background

Generic medicines are similar to innovator medicine in terms of safety, quality, efficacy, dosage form, strength, and route of administration. They have the same therapeutic use to innovator medicines and available at a far lower price. However, health professionals' poor knowledge and attitude may limit its utilization. The present study aimed at assessing the knowledge, attitude, and practice of pharmacy professionals towards generic medicines in Harar city, Eastern Ethiopia.

### Methods

A cross-sectional survey was conducted among community pharmacists in Harar city. A self-administered thirty-three item questionnaire on Knowledge, attitude, and practice of community pharmacists was utilized. Logistic regression analysis was performed to predict the determinants of knowledge and attitude of pharmacists. An odds ratio at 95% confidence interval along with a p-value < 0.05 was considered significant.

### Results

Among 80 community pharmacists' approached, 74 completed the survey, providing a response rate of 92.5%. Sixty-seven percent of the respondents knew that generic drugs are bioequivalent to brand drugs and claimed generic medicines are cheaper (86.5%). Nearly half (48.6%) of participants believe that generic medicines are less effective and slower in the onset of action (58.1%). More than half (54.1%) of study participants revealed their lack of belief in generic medicine as a factor hindering the selection and dispensing of generic medicines. In multivariate logistic regression, experience in community pharmacy practice (Adjusted odds ratio (AOR = 2.18, 95%CI: 1.21–63.1) and Sex (AOR = 3.88, 95% CI: 2.12–39.62) were significantly associated with knowledge and attitude toward generic medicines, respectively.

**Data Availability Statement:** All relevant data are within the paper and its Supporting Information files.

**Funding:** The author(s) received no specific funding for this work.

**Competing interests:** the authors have declared that no competing interests exist.

**Abbreviations:** EPA, Ethiopian pharmaceutical association; CDROs, community drug retail outlets; CI, Confidence Interval; SPSS, Statistical Packages for Social Sciences.

## Conclusion

The current study revealed that there is a gap in the knowledge and attitude of community pharmacists towards generic and brand drugs. More than averages of the respondents have known the concept of generic medicine including their right to perform generic substitution and had a positive attitude toward generics. Female pharmacists were more likely to have a positive attitude and the overall knowledge was higher in those who have more than 5 years of work experience.

## Introduction

Pharmacy practice system is the responsible provision of drug therapy to achieve a definitive outcome that improves patients' quality of life. The role of the pharmacist has evolved substantially in recent decades from traditional roles of compounding and dispensing of medication to providing direct patient care to ensure the rational and cost-effective use of medicines, including the promotion of equally effective and less expensive generic medicines utilization [1].

A generic drug is defined as "a pharmaceutical product, usually intended to be interchangeable with the branded medicine. It can only be manufactured after a patent protection expires of the branded medicine. Generics medicines are also considered as therapeutically equivalent to their counterpart branded drugs" [2].

Generic drugs become available after patent protection for branded drugs has expired. After the expiry of patent protection, the generic drugs could be manufactured by companies other than the innovator company [3]. Generic medicine is the same as its comparator branded medicine in terms of safety, quality, efficacy, dosage form, strength, route of administration, and intended use [4–7]. Nevertheless, they may be different in some other aspects including shape and packaging [8].

Generic medicines are typically 20 to 90% cheaper than their brand equivalent. Since the cost for drug discovery, pre-clinical and clinical trials, as well as for some other reasons will not be considered and patent rights will no longer be protected, generics usually sold at a far lower price. Hence, generic medicine is an economical alternative to more expensive branded medicine and in reducing pharmaceutical monetary expenditures [9, 10]. Generic substitution is considered as cost minimization strategy, by government and other third-party payers, to contain pharmaceutical expenditure [11–15].

Currently, several countries around the world advocate generic utilization by instigating different policies, initiatives, and strategies. Many countries allow or authorize pharmacists to substitute a generic version (if one is available) whenever a physician prescribed. This flexibility is imperative to promote generic medicines utilization [16–21].

In Ethiopia, the use of generic names of drugs in prescription is enforced by clinical practice guidelines [22]. Besides, the pharmacist has the right to dispense generic drugs as substitutes for prescribed branded drugs as on national drug policy [23].

Generic prescribing and generic substitution have been controversial. Furthermore, the acceptance and promotion of generic medicines among healthcare professionals has been the main concern [24–26]. Issues related to quality, safety, and efficacies of the generic medicines are of top of concerns [25, 27, 28]. Misinformation or inadequate knowledge of healthcare professionals regarding generics causes' hesitation on the utilization of generics and it has been

the central challenge against wider use of these products [29]. Thus, it is crucial to analyze the knowledge and attitude of health professionals regarding generic medicines which is a prerequisite to encourage its utilization [6, 28].

Pharmacists are the main determinants in consumer choice to the rational use of generics, according to a study on consumers' perception [30]. A previous study in Ethiopia employed pharmacists has revealed that more than half (52.9%) of participants had known that generic medicine is bioequivalent to branded medicine and 34.4% of respondents have believed generic medicines are less effective [31]. This research can provide evidence of the community pharmacists' knowledge, attitude and practice toward generic medicines and to identify factors associated with generic medicines dispensing in Ethiopia. The findings may assist policy-makers regarding generic substitution in the future.

## Materials and methods

### Study design and setting

In this cross-sectional survey conducted on the facility-based census, a questionnaire was distributed to all community pharmacists in Harar city from September to October 2018. The study was conducted in all community drug retail outlets (CDROs) in Harar City. The city is located in Harari regional state, which is about 526 km away in the Eastern direction from Addis Ababa, the capital of Ethiopia. According to the report from the 2007 national population and housing census, the city had an estimated total population of 183,344 (ECSA, 2007). Harar city has 45 drug stores and 16 community pharmacies.

**Sample size.** All community drug retail outlets in Harar were considered and all pharmacy professionals working in CDROs were included. Accordingly, there are 61 community drug retail outlets in Harar city and approximately around 90 pharmacy professionals are working in community drug retail outlets. Among those, 80 of them were volunteered to participate in the study.

### The study tool

A self-administered thirty three-item questionnaire on Knowledge, attitude, and practice of community pharmacists toward generic medication was developed after an extensive review of available literature related to the topic [27, 28, 30]. The questionnaire was prepared in English. This was translated to Amharic followed by back translation to English to ensure that the translated version gives the right meaning. The tool has two-part with the first part contained socio-demographic information of study participants. Part two questions were to assess their knowledge, attitude and practice regarding generic medicines and were assessed using the 5-point Likert scale (1 = strongly agree to 5 = strongly disagree). Questions assessing the knowledge had only one "correct" answer and each correct answer worth one point. The total knowledge score which ranged from 8–40 was used to classify knowledge into adequate knowledge (24–40) and inadequate knowledge which is a score of less than the average score (8–23). Question on attitude had 9–45 total score that used to delineate between negative and positive attitude while 27 took as cut off point. Negatively worded questions were reverse scored during the analysis so that the higher scores reflected a more positive attitude towards generic medicines. The questionnaire was pretested on 5% of community pharmacists in Gurawa town, which is a town near Harar city and necessary modifications were instituted to final data collection tool depending on the feedback obtained from the pilot study. Data were collected by three pharmacy professionals and adequate training was provided to data collectors on contents of the questionnaire, data collection methods and ethical concerns by the principal investigator. Questionnaires were collected three days from the date of distribution.

## Data analysis

The responses from each study participants were manually entered into Statistical Package for Social Studies (SPSS) version 21.0 and were double-checked for accuracy. Descriptive aspects of analyses were done through calculating mean, standard deviation, frequencies, and percentages. Bivariate logistic regression analysis was done to predict the determinants of knowledge and attitude towards generic medicines. Variables with a p-value of less than 0.2 were included in the multivariate logistic regression analysis. Odds ratio with 95% confidence interval (95% CI) were also computed along with corresponding $p$-value ($p<0.05$).

## Ethical considerations

Ethical clearance was secured from the institutional research board, School of Pharmacy at Haramaya University. The purpose and importance of the study were explained to and written consent was obtained from each study participant before proceeding to query. Furthermore, other issues relating to the ethical conduct of research: such as confidentiality, privacy was upheld.

## Result

### Socio-demographic and work profiles

Out of 80 community pharmacists' approached, 74 completed the survey (response rate-92.5%). More than half (55.4%) participants were female and within age group of 20–29 years (54.1%). Around two-third (63.5%) of the respondents held diploma in pharmacy and worked as an employee in community pharmacy (60.8%). Sixty-one (82.4%) of the respondents were not Ethiopian Pharmacists Association (EPA) member and only about one-fourth of participants (24.3%) had above 5 years' experience in community pharmacy practice (Table 1).

### Pharmacy professionals' knowledge on generic medicines

In our study, majority (86.5%) of participants knew that generic medicines are cheaper than branded medicines and they are bioequivalent to branded medicines (67.6%). On the other hand, more than one-third (39.2%) of participants did not support the idea that extensive use of generic medicines in Ethiopia helps in minimizing the country's health care expenditure (Table 2).

### Knowledge of community pharmacists and associated factors

In multivariate logistic regression, only work experience has shown statistically significant association with knowledge level. Pharmacists with 5 or more years' experience in community pharmacy practice were more likely to have adequate knowledge (AOR = 2.18, 95%CI:1.21–63.1). The remained socio-demographic and work profiles were not significantly associated with knowledge level of respondents, both in bivariate and multivariate logistic regression analysis (Table 3).

### Pharmacists' attitudes toward generic medicines

Among study participants, around half (48.6%) believed that generic medicines are less effective than brand medicines and more than average (56.8%) revealed that branded medicines are of higher quality. Thirty-five (41.9%) participants disagreed generics has a slower onset of action and fifty percent of respondents cited the differences in prices between branded and generic versions as a reason for generic substitution (Table 4).

**Pharmacists' attitude and associated factors.** In this study, female pharmacy professionals were more likely to have a positive attitude toward generic medicines in multivariate logistic regression (AOR = 3.88, CI: 2.12–39.623). The remained socio-demographic and work

**Table 1. Socio-demographic characteristics of the community pharmacists in Harar, Eastern Ethiopia 2018.**

| Variables | Frequency (%) |
|---|---|
| **Age** | |
| 20–29 years | 40(54.1) |
| 30–39 years | 26(35.1) |
| 40 and above years | 8(10.8) |
| **Sex** | |
| Male | 33(44.6) |
| Female | 41(55.4) |
| **Qualification** | |
| Diploma | 47(63.5) |
| Degree | 21(28.4) |
| MSc | 6(8.1) |
| **Employment position** | |
| Pharmacy /drug store owner | 26(35.1) |
| Employee | 45(60.8) |
| Others | 3(4.1) |
| **Education Institution** | |
| Private institution | 48(64.9) |
| Government/public institution | 26(35.1) |
| **Ethiopian pharmacists association (EPA) membership** | |
| Yes | 13(17.6) |
| No | 61(82.4) |
| **Work experience** | |
| Less than 2 years | 29(39.2) |
| 2–5 years | 27(36.5) |
| Greater than 5 years | 18(24.3) |

profiles have not revealed a significant association with pharmacists' attitude towards generic medicines (Table 5).

**Pharmacy professionals' views toward locally manufactured generic medicines.** In present study, forty-six percent (agree = 20.3%, strongly agree 25.7%) of the respondents claimed that locally manufactured generics are similar in quality to the imported generics and refused the idea that manufacturers of local generic products have a reliable logistic and

**Table 2. Knowledge of community pharmacists about generic medicine in Harar, Eastern Ethiopia 2018.**

| Variables | Strongly disagree (%) | Disagree (%) | Neutral (%) | Agree (%) | Strongly agree (%) |
|---|---|---|---|---|---|
| A generic medicine is bioequivalent to branded medicine | 10(13.5) | 5(6.8) | 9(12.2) | 29(39.2) | 21(28.3) |
| generic medicine must contain the same amount of active ingredients as the branded medicine | 6(8.1) | 8(10.8) | 6(8.1) | 35(47.3) | 19(25.7) |
| A generic medicine must be in the same dosage form to the branded medicine | 5(6.8) | 14(18.9) | 11(14.9) | 26(35.1) | 18(24.3) |
| Generic medicines are cheaper than branded medicines | 4(5.4) | 4(5.4) | 2(2.7) | 43(58.1) | 21(28.4) |
| Wider use of generic medicines in Ethiopia helps in decreasing the country's health care expenditure | 17(23) | 12(16.2) | 13(17.5) | 15(20.3) | 17(23) |
| Community pharmacists in Ethiopia have the right to perform generic substitution | 8(10.8) | 9(12.2) | 7(9.5) | 32(43.2) | 18(24.3) |
| Generic substitution of medicines with narrow therapeutic index is inappropriate | 9(12.2) | 8(10.8) | 4(5.4) | 32(43.2) | 21(28.4) |
| In Ethiopia, generic medicines are approved by FMHACA as brand medicines | 6(8.1) | 8(10.8) | 7(9.5) | 35(47.3) | 18(24.3) |

**Table 3. Predictors of knowledge level of community pharmacists on generic medicines in Harar, Eastern Ethiopia 2018.**

| | Knowledge on generics | | | |
|---|---|---|---|---|
| Variables | Adequate [%] | Inadequate [%] | Crude OR [95%CI] | Adjusted OR [95%CI] |
| **Sex** | | | | |
| Male | 14 | 19 | 1 | 1 |
| Female | 22 | 19 | 1.57(0.53–1.603) | 0.46(0.29–3.951) |
| **Age** | | | | |
| 20–29 years | 19 | 21 | 1 | 1 |
| 30–39 years | 15 | 11 | 1.507(0.66–20.5) | 0.297(0.014–6.078) |
| 40 and above years | 2 | 6 | 2.71(0.41–1.449) | 0.520(0.06–44.7) |
| **Qualification** | | | | |
| Diploma | 13 | 25 | 1 | 1 |
| Degree | 9 | 11 | 1.57(0.363–13.01) | 0.689(0.3–14.7) |
| MSc | 4 | 2 | 3.84(0.61–16.547) | 7.044(0.35–140.01) |
| **Employment position** | | | | |
| Partime /full-time owner | 5 | 21 | 1 | 1 |
| Employee | 30 | 15 | 0.11(0.057–28.02) | 0.87(0.547–5.94) |
| Others | 1 | 2 | 2.1(0.02–2.983) | 1.82(0.12–61.23) |
| **Education Institution** | | | | |
| Private institution | 34 | 14 | 1 | 1 |
| Government/public institution | 24 | 2 | 4.94(0.318–8.922) | 7.1(0.8–23.21) |
| **EPA membership** | | | | |
| Yes | 6 | 7 | 1.35(0.267–2.942) | 1.7(0.057–50.729) |
| No | 31 | 30 | 1 | 1 |
| **Work experience** | | | | |
| Less than 2 years | 19 | 10 | 1 | 1 |
| 2–5 years | 5 | 22 | 8.36(0.657–21.74) | 5.87(0.054–41.54) |
| Greater than 5 years | 12 | 6 | 1.052(1.012–20.085)* | 2.18(1.21–63.1)* |

supply system (44.6%). From respondents, more than half (56.7%) of believed that locally manufactured generics are equal in safety and efficacy compared to the imported generics and claimed that locally manufactured generics are cheaper (77%) [Table 6].

**Table 4. An attitude of community pharmacists towards generic medicines in Harar, Eastern Ethiopia 2018.**

| Variables | Strongly disagree (%) | Disagree (%) | Neutral (%) | Agree (%) | Strongly agree (%) |
|---|---|---|---|---|---|
| Generic medicines are less effective than branded medicines | 6(8.1) | 23(31.1) | 9(12.2) | 26(35.1) | 10(13.4) |
| Branded medicines are of higher quality compared to generic drugs | 6(8.1) | 18(24.3) | 8(10.8) | 23(31.1) | 19(25.7) |
| Generic drugs produce more side effects than brand name medicines | 23(31.1) | 22(29.7) | 9(12.2) | 12(16.2) | 8(10.8) |
| Generic medicine has slow onset of action | 18(24.3) | 13(17.6) | 5(6.8) | 30(40.5) | 8(10.8) |
| I support generic substitution for branded medicines in all cases where a generic is available | 14(18.9) | 7(9.5) | 17(23) | 19(25.6) | 17(23) |
| The price difference between generic and branded medicine would be good reason to dispense generics especially for people who do not have prescription medicine benefits in Ethiopia | 8(10.8) | 16(21.6) | 13(17.6) | 33(44.6) | 4(5.4) |
| Patients should be briefed on the reasons for choosing generic medicines | 9(12.2) | 3(4.1) | 10(13.5) | 38(51.4) | 14(18.8) |
| The intensity of promotional activities by medical representatives is crucial for dispensing generics | 5(6.8) | 15(20.3) | 6(8.1) | 23(31.1) | 25(33.7) |
| Community pharmacists should be allowed to perform generic substitutions without consulting prescribing physicians | 11(14.9) | 15(20.3) | 10(13.5) | 30(40.5) | 8(10.8) |

**Table 5. Predictors of attitude of community pharmacists towards generic medicines, Harar, Ethiopia, 2018.**

| Variables | Good | Poor | Crude OR [95%CI] | Adjusted OR [95%CI] |
|---|---|---|---|---|
| **Attitude on generics** | | | | |
| **Sex** | | | | |
| Male | 15 | 18 | 1 | 1 |
| Female | 34 | 7 | 5.829(2.012–16.884)* | 3.88(2.12–39.623)* |
| **Age** | | | | |
| 20–29 years | 34 | 10 | 1 | 1 |
| 30–39 years | 10 | 12 | 4.08(0.099–2.417) | 0.686(0.071–6.636) |
| 40 and above years | 5 | 3 | 2.04(0.381–10.511) | 0.855(0.075–9.756) |
| **Qualification** | | | | |
| Diploma | 31 | 17 | 1 | 1 |
| Degree | 13 | 7 | 1.018(0.296–25.424) | 7.805(0.434–38.219) |
| MSc | 5 | 1 | 2.74(0.261–27.821) | 2.453(0.116–52.063) |
| **Employment position** | | | | |
| Partime /full-time owner | 16 | 10 | 1 | 1 |
| Employee | 31 | 14 | 0.72(0.1–15.647) | 0.287(0.04–19.534) |
| Others | 2 | 1 | 0.80(0.075–10.808) | 0.88(0.015–53.059) |
| **Education Institution** | | | | |
| Private institution | 34 | 14 | 1 | 1 |
| Government/public institution | 15 | 11 | 1.78(0.207–1.521) | 4.627(0.126–169.57) |
| **EPA membership** | | | | |
| Yes | 7 | 6 | 1 | 1 |
| No | 42 | 19 | 1.895(0.561–6.403) | 0.460(0.02–9.710) |
| **Work experience** | | | | |
| Less than 2 years | 19 | 10 | 1 | 1 |
| 2–5 years | 15 | 12 | 1.52(0.613–11.298) | 6.1(0.73–23.190) |
| More than 5 years | 15 | 3 | 3.1(0.935–17.113) | 0.591(0.324–76.92) |
| **Knowledge on generics** | | | | |
| No | 23 | 15 | 1 | 1 |
| Yes | 26 | 10 | 0.59(0.222–1.567) | 0.4(0.073–2.182) |

**Table 6. View of community pharmacists towards locally manufactured medicines in Harar, Ethiopia 2018.**

| Variables | Strongly disagree (%) | Disagree (%) | Neutral (%) | Agree (%) | Strongly Agree (%) |
|---|---|---|---|---|---|
| Locally manufactured generics are equal in quality compared to the imported generics | 7(9.5) | 16(21.6) | 17(23) | 15(20.3) | 19(25.7) |
| Locally manufactured generics are equal in safety and efficacy to the imported generics | 11(14.9) | 8(10.8) | 13(17.6) | 26(35.1) | 16(21.6) |
| Manufacturers of local generic products have a reliable logistic and supply system | 21(28.4) | 12(16.2) | 7(9.5) | 13(17.6) | 21(28.4) |
| I prefer to stock and dispense locally manufactured generics because of the financial incentives provided by companies | 19(25.7) | 26(35.1) | 15(20.3) | 7(9.5) | 7(9.5) |
| Credibility of the generic manufactures/suppliers is my concern when stocking medicines in my pharmacy | 15(20.3) | 13(17.6) | 12(16.2) | 22(29.7) | 12(16.2) |
| I will only stock locally manufactured product which is well advertised by the company. | 15(20.3) | 18(24.3) | 13(17.6) | 12(16.2) | 16(21.6) |
| Imported generics need to pass a more stringent approval process compared with locally manufactured ones. | 8(10.8) | 13(17.6) | 14(18.90) | 30(40.5) | 9(12.2) |
| Locally manufactured generics are cheaper compared to imported generics. | 6(8.1) | 7(9.5) | 4(5.4) | 26(35.1) | 31(41.9) |
| Drug Regulatory Authorities need to convince pharmacists on higher quality of locally manufactured generics | 7(9.5) | 15(20.3) | 10(13.5) | 29(39.2) | 13(17.6) |

**Table 7. Possible factors to influence selection and dispensing of generic medicines among the community pharmacists in Harar, Ethiopia 2018.**

| Variables | Not important (%) | Neutral (%) | Important (%) |
|---|---|---|---|
| Lack of belief in generic medicines | 17(23) | 17(23) | 40(54) |
| Availability of policies, laws & regulations | 25(33.8) | 9(12.2) | 40(54) |
| Affordability to the customer | 9(12.2) | 7(9.5) | 58(78.3) |
| Lacking options | 44(59.5) | 5(6.8) | 25(33.7) |
| Consumer preference/ demand | 99(25.7) | 18(24.3) | 37(50) |
| Cost-effectiveness of generic medicines | 6(8.1) | 4(5.4) | 64(86.5) |
| Substitution agreement with the prescriber | 19(25.7) | 9(12.2) | 46(62.1) |

### Influencing factors related to selection and dispensing of generic medicines

In this study, more than half (54%) of study participants revealed their lack of belief in generic medicine as an important factor that deters selection and dispensing of generic medicines. From participants, fifty-eight (78.3%) and sixty-four (86.5%) of them declared affordability to customer and cost-effectiveness of generic medicines as determinant factors for selection and dispensing of generic medicines, respectively [Table 7].

## Discussion

This study tried to assess the knowledge, attitude, and practice of community pharmacists toward generic medicines and also their views toward locally produced generic medicine in Ethiopia. The finding this study shows that there are gaps in knowledge, attitude, and practice of community pharmacy professionals towards generic medicines, which is corroborated by the previous studies [30, 31]. This could be explained by pharmacy professionals' individualized knowledge and attitudes towards generic medicines.

In this study, fifty (67.6%) of the respondents agreed that generic medicines are bioequivalent to their counterpart branded medicines, which is likely to a study finding in New Zealand (70%) [25]. However, it was higher compared to findings from a previous studies conducted in Northern Ethiopia (52.9%) [31] and Malaysia (50.2%) [32].

In this study, sixty-four (86.5%) of study subjects claimed that generic medicines are cheaper than branded medicines, which is comparable with the finding of a study conducted in Australia (91.3%) [33]. In addition, more than half (59.4%) of the participants knew that generic medicines must be in the same dosage form as the branded medicines, which is comparable with a study conducted by Yared et al. (64.3%) [31]. In contrast, it was much lower as compared to a study in Australia (84.1%) [33]. This could be due to educational variance between respondents employed in two studies.

Our study revealed that most (67.5%) of pharmacists know their right to perform the generic substitution. This was much lower than Allenet et al.'s study that announced 90% of the pharmacists were agreed with pharmacists' right of generic medicines substitution [34]. This could be explained by the better progress of pharmacy care practice in France. In multivariate logistic regression, only work experience had a statically significant association with knowledge level of respondents and it showed that pharmacists with 5 or more years of experience were more likely to have adequate knowledge (AOR = 2.18, 95% CI: 1.21–63.1). This could be due to the fact, as the number of years of practice increases, pharmacists are more likely to be exposed to different local generics manufacturing pharmaceutical companies and local health authorities' provided information on generic medicines.

In this study, more than one-third (39.2%) of respondents were disagreed with the myth that generic medicines are less effective than branded medicines, which was much lower than

reports of Yared et al. (50%) [31] and Hassali et al. (58.4%) [32]. This implicates the better perception of generic medicines in Malaysia and even in Northern Ethiopia than the study area, which indicates the need to promote generic medicines in Harar city.

In our study, more than half (56.8%) of respondents support the concept of branded medicines are of higher quality, which is comparable to a study report from Mekelle (51.7%) [31], but far lower than in Australian comparative study of senior medical students and pre-registrant pharmacists (89.6%) [33]. This might be due to the inclusion of both junior and senior pharmacists in our study, unlike Australian study which employed only pre-registrants pharmacists. Nevertheless, the finding is much lower than Othman et al.'s study on pharmacy students in Yemeni, in which 98% and 96% of pharmacists accepted the idea that generics are similar in quality and as effective as branded medicines, respectively [35]. The probable explanation to this discrepancy might be due to students' familiarity to basic science than working pharmacy professionals.

In current study, around half (51%) of study participants stated that pharmacists should be allowed to make a generic substitution without consulting physicians. This finding is comparable to a previous study report (50.5%) [31]. In contrast, it was lower than a study in Turkey in which 55% of the respondents claimed that pharmacists may substitute a generic without consulting the prescribers [36].

Among study participants, forty (54.1%) of them claimed their lack of belief in generic medicines as a determinant factor that affects the dispensing of generic medicines. This is higher compared with a previous study report from Northern Ethiopia 48.3% [31]. This pharmacists' high level of lack of belief on generics indicates that correcting inappropriate belief of pharmacy professionals toward generic medicines should be the main concern for all stakeholder including pharmaceutical association in the country, to ensure generic dispensing practice in Ethiopia.

From study participants, sixty-four (86.5%) of them declared cost-effectiveness of generic medicines as a significant factor in dispensing of a generic products. This was higher than Yared et al.'s study in northern Ethiopia (73.6%) [31]. However, it was lower than the finding of a study conducted in Australia (91.3%) [33]. The discrepancy with Australia could be due to the sustainability of generic medicines which decrease the price of generic medicine in Australia.

Furthermore, thirty-four (46%) of study participants revealed that locally manufactured generics are equal in quality to the imported generics. This is comparable with Bashaar et al.'s study in which 47.5% of participants have supported the concept of locally manufactured generics are equal in quality to imported generics [37]. However, it is lower compared to a finding of a study conducted in Mekelle [31], which could be due to the presence of local generic manufacturing pharmaceutical companies and associated exposure to generic related information in northern Ethiopia. Thus, recommended to local generic manufacturing pharmaceutical companies to address their promotion to the area with no pharmaceutical company including the Eastern part of the country.

In the present study, fifty-seven (77%) of the respondents accepted the idea that locally manufactured generics are cheaper than imported generics and thirty-two (56.7%) of them agreed that locally manufactured generics are equal in safety and efficacy to imported generics. This finding is lower than a report from a study performed in Malaysia (58.4%) [32]. Future studies are warranted to explore the potential intervention to address the gaps in knowledge, attitude, and practice of pharmacy professionals on generic medicines. Finally, qualitative research on the area with in-depth interviews is required to comprehend the real-life barriers to dispense generics that the pharmacists are facing in Ethiopia.

## Conclusions

The current study revealed that there is a gap in the knowledge, attitude, and practice of community pharmacists towards generic and brand drugs. Despite this, more than averages of the respondents were knowledgeable on the concept of generic medicine, including their right to perform generic substitution and had a positive attitude toward generics. In multivariate analysis, female pharmacists were more likely to have a positive attitude and overall knowledge was higher in those who have more than 5 years of work experience. More than half of the study participants claimed the lack of belief in generic medicines as significant factors that affect the selection and dispensing of generics.

## Supporting information

**S1 File.**
(DOCX)

**S1 Dataset.**
(ZIP)

## Acknowledgments

Acknowledgments
We are grateful to all participants of the study.

## Author Contributions

**Conceptualization:** Ammas Siraj Mohammed, Fuad Adem Mohammed.

**Data curation:** Fuad Adem Mohammed.

**Formal analysis:** Ammas Siraj Mohammed, Nigist Alemayehu Woldekidan, Fuad Adem Mohammed.

**Methodology:** Ammas Siraj Mohammed, Nigist Alemayehu Woldekidan, Fuad Adem Mohammed.

**Supervision:** Ammas Siraj Mohammed.

**Validation:** Ammas Siraj Mohammed.

**Writing – original draft:** Ammas Siraj Mohammed, Fuad Adem Mohammed.

**Writing – review & editing:** Ammas Siraj Mohammed, Nigist Alemayehu Woldekidan, Fuad Adem Mohammed.

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
