## [Decision Letter · Decision Letter 0]

19 Feb 2020

PONE-D-19-29738

Knowledge, Attitude and Practice of Pharmacy Professionals towards Generic Medicines, Eastern Ethiopia: A Cross-Sectional Study

PLOS ONE

Dear mr Mohammed,

Thank you for submitting your manuscript to PLOS ONE. After careful consideration, we feel that it has merit but does not fully meet PLOS ONE’s publication criteria as it currently stands. Therefore, we invite you to submit a revised version of the manuscript that addresses the points raised during the review process.

We would appreciate receiving your revised manuscript by Apr 04 2020 11:59PM. To enhance the reproducibility of your results, we recommend that if applicable you deposit your laboratory protocols in protocols.io, where a protocol can be assigned its own identifier (DOI) such that it can be cited independently in the future. For instructions see: http://journals.plos.org/plosone/s/submission-guidelines#loc-laboratory-protocols

We look forward to receiving your revised manuscript.

Kind regards,

Joseph Telfair, DrPH, MSW, MPH

Academic Editor

PLOS ONE

Journal Requirements:

2. Please provide additional details regarding participant consent. In the ethics statement in the Methods and online submission information, please ensure that you have specified whether consent was written or verbal/oral. If consent was verbal/oral, please specify: 1) whether the ethics committee approved the verbal/oral consent procedure, 2) why written consent could not be obtained, and 3) how verbal/oral consent was recorded. If your study included minors, please state whether you obtained consent from parents or guardians in these cases.

3. Please include copies of the survey questions or questionnaires used in the study, in both the original language and English, as Supporting Information, or include a citation if they have been published previously.

4. We noticed you have some minor occurrence of overlapping text with the following previous publication(s), which needs to be addressed:

https://www.jbclinpharm.org/articles/assessment-of-knowledge-attitude-and-practice-of-pharmacy-professionals-toward-generic-medicines-northern-ethiopia-mekelle-a-cross-3923.html

In your revision ensure you cite all your sources (including your own works), and quote or rephrase any duplicated text outside the Methods section. Further consideration is dependent on these concerns being addressed.

5. We suggest you thoroughly copyedit your manuscript for language usage, spelling, and grammar. If you do not know anyone who can help you do this, you may wish to consider employing a professional scientific editing service.  

6. Your ethics statement must appear in the Methods section of your manuscript. If your ethics statement is written in any section besides the Methods, please move it to the Methods section and delete it from any other section. Please also ensure that your ethics statement is included in your manuscript, as the ethics section of your online submission will not be published alongside your manuscript.

7. We note you have included a table to which you do not refer in the text of your manuscript. Please ensure that you refer to Table 2 in your text; if accepted, production will need this reference to link the reader to the Table.

Reviewers' comments:

Reviewer's Responses to Questions

**Comments to the Author**

1. Is the manuscript technically sound, and do the data support the conclusions?

Reviewer #1: Yes

2. Has the statistical analysis been performed appropriately and rigorously? 

Reviewer #1: Yes

3. Have the authors made all data underlying the findings in their manuscript fully available?

Reviewer #1: Yes

4. Is the manuscript presented in an intelligible fashion and written in standard English?

Reviewer #1: No

5. Review Comments to the Author

Reviewer #1: This is an interesting study to uncover the knowledge, beliefs and attitudes among pharmacists in Ethiopia and other related professionals, about the value of generic drugs. The approach and methods are well described. The authors should describe the pharmacy practice system at the beginning, so that the results are put in the context of pharmacy practice and can be better interpreted.

In the discussion, there should also be a more thorough explanation of the 'so what' and the next steps.

In general, there are many grammatical and typographical errors that needs to be corrected, and the formatting should be improved.

6. PLOS authors have the option to publish the peer review history of their article (what does this mean?). If published, this will include your full peer review and any attached files.

Reviewer #1: No

---

## [Author Response · Author response to Decision Letter 0]

13 Apr 2020

Editor(s)’ decision and comments

Thank you for submitting your manuscript to PLOS ONE. After careful consideration, we feel that it has merit but does not fully meet PLOS ONE’s publication criteria as it currently stands. Therefore, we invite you to submit a revised version of the manuscript that addresses the points raised during the review process.

 Our answer: thank you for your consideration of its merit and offer.

Our answer: thank you. No change in financial disclosure

To enhance the reproducibility of your results, we recommend that if applicable you deposit your laboratory protocols in protocols.io, where a protocol can be assigned its own identifier (DOI) such that it can be cited independently in the future. For instructions see: http://journals.plos.org/plosone/s/submission-guidelines#loc-laboratory-protocols

 Our answer: thank you. It is not applicable 

Response to editor

 Our Answer: Thank you for this comment. The revised manuscript version is formatted in strict adherence to PLOS ONE’s formatting guideline, which obtained from above URL address, including file naming, font size for each section, spacing and other required format.

2. Please provide additional details regarding participant consent. In the ethics statement in the Methods and online submission information, please ensure that you have specified whether consent was written or verbal/oral. If consent was verbal/oral, please specify: 1) whether the ethics committee approved the verbal/oral consent procedure, 2) why written consent could not be obtained, and 3) how verbal/oral consent was recorded. If your study included minors, please state whether you obtained consent from parents or guardians in these cases.

 Our answer: Thank you for rising this point. Since the study participants are expected to read and write, written informed consent was obtained from all subjects before proceeding to query, which was approved by ethical review committee. During the informed consent process, study participants were assured that data collected would be used only for stated purposes and would not be disclosed or released to others without the consent of the participants. All study participants was adults, pharmacy professionals. 

3. Please include copies of the survey questions or questionnaires used in the study, in both the original language and English, as Supporting Information, or include a citation if they have been published previously.

 Our answer: Thank you for this comment. The questionnaire used in this study is not published anywhere, but the literature we derived it from were adequately cited and it is provided as supplementary material both in local language and English upon submission.

4. We noticed you have some minor occurrence of overlapping text with the following previous publication(s), which needs to be addressed:

https://www.jbclinpharm.org/articles/assessment-of-knowledge-attitude-and-practice-of-pharmacy-professionals-toward-generic-medicines-northern-ethiopia-mekelle-a-cross-3923.html. In your revision ensure you cite all your sources (including your own works), and quote or rephrase any duplicated text outside the Methods section. Further consideration is dependent on these concerns being addressed.

 Our answer: thank you for rising this point. We have checked the manuscript for any possible overlapping text to different source including the one you have indicated above and the obtained overlapping text were rephrased and synthesized using authors’ specific language

5. We suggest you thoroughly copyedit your manuscript for language usage, spelling, and grammar. If you do not know anyone who can help you do this, you may wish to consider employing a professional scientific editing service. 

 Our answer: thank you for this comment. After copyediting the manuscript thoroughly, we have tried to address all typographical and grammatical errors in the manuscript. The grammatical and typographical errors within manuscript was addressed in collaboration with my colleagues 

6. Your ethics statement must appear in the Methods section of your manuscript. If your ethics statement is written in any section besides the Methods, please move it to the Methods section and delete it from any other section. Please also ensure that your ethics statement is included in your manuscript, as the ethics section of your online submission will not be published alongside your manuscript.

 Our answer: we have moved the ethics statement from declaration section to method section of the manuscript. 

7. We note you have included a table to which you do not refer in the text of your manuscript. Please ensure that you refer to Table 2 in your text; if accepted, production will need this reference to link the reader to the Table.

 Our answer: we have included citation of table2 in the text prior to the table as per your recommendation.

Reviewers comment and decision

1. Is the manuscript technically sound, and do the data support the conclusions?

Reviewer #1: Yes

2. Has the statistical analysis been performed appropriately and rigorously? 

Reviewer #1: Yes

3. Have the authors made all data underlying the findings in their manuscript fully available?

Reviewer #1: Yes

4. Is the manuscript presented in an intelligible fashion and written in standard English?

 Response to reviewer

 Reviewer #1: This is an interesting study to uncover the knowledge, beliefs and attitudes among pharmacists in Ethiopia and other related professionals, about the value of generic drugs. The approach and methods are well described. The authors should describe the pharmacy practice system at the beginning, so that the results are put in the context of pharmacy practice and can be better interpreted.

In the discussion, there should also be a more thorough explanation of the 'so what' and the next steps.

In general, there are many grammatical and typographical errors that needs to be corrected, and the formatting should be improved

 Our answer: thank you for your appreciation and offer. Pharmacy practice system is described at the beginning of manuscript including its relation with generic drug utilization and we have tried to put and interpret the result in context of pharmacy practice system. In discussion, we have included that what should be done per our each finding and at the end of discussion, future direction for studies that will be conducted on the area. After looking the manuscript thoroughly, we have tried to address all grammatical and typographical errors in manuscript and the revised version of manuscript is formatted in strict adherence to PLOS ONE’s formatting style including naming of file.

---

## [Editor Report · Decision Letter 1]

11 Jun 2020

Knowledge, Attitude and Practice of Pharmacy Professionals on Generic Medicines in Eastern Ethiopia: A Cross-Sectional Study

PONE-D-19-29738R1

Dear Dr. Mohammed,

We’re pleased to inform you that your manuscript has been judged scientifically suitable for publication and will be formally accepted for publication once it meets all outstanding technical requirements.

Kind regards,

Joseph Telfair, DrPH, MSW, MPH

Academic Editor

PLOS ONE
---

## [Editor Report · Acceptance letter]

1 Jul 2020

PONE-D-19-29738R1 

Knowledge, attitude, and practice of pharmacy professionals on generic medicines in Eastern Ethiopia: A cross-Sectional Study 

Dear Dr. Mohammed:

I'm pleased to inform you that your manuscript has been deemed suitable for publication in PLOS ONE. Congratulations! Your manuscript is now with our production department. 

Kind regards, 

on behalf of

Dr. Joseph Telfair 

Academic Editor

PLOS ONE